# Colocalization of MID1IP1 and c-Myc is Critically Involved in Liver Cancer Growth via Regulation of Ribosomal Protein L5 and L11 and CNOT2

**DOI:** 10.3390/cells9040985

**Published:** 2020-04-16

**Authors:** Ji Hoon Jung, Hyo-Jung Lee, Ju-Ha Kim, Deok Yong Sim, Eunji Im, Sinae Kim, Suhwan Chang, Sung-Hoon Kim

**Affiliations:** 1Cancer Molecular Targeted Herbal Research Laboratory, College of Korean Medicine, Kyung Hee University, 26 Kyungheedae-ro, Dongdaemun-gu, Seoul 02447, Korea; johnsperfume@gmail.com (J.H.J.); hyonice77@naver.com (H.-J.L.); 964juha@daum.net (J.-H.K.); simdy0821@naver.com (D.Y.S.); ji4137@naver.com (E.I.); 2Department of Biomedical Sciences, University of Ulsan, College of Medicine, Asan Medical Center, Seoul 05505, Korea; skim368@gmail.com (S.K.); suhwan.chang@amc.seoul.kr (S.C.)

**Keywords:** MID1IP1, CNOT2, RPL5, RPL11, c-Myc, colocalization

## Abstract

Though midline1 interacting protein 1 (MID1IP1) was known as one of the glucose-responsive genes regulated by carbohydrate response element binding protein (ChREBP), the underlying mechanisms for its oncogenic role were never explored. Thus, in the present study, the underlying molecular mechanism of MID1P1 was elucidated mainly in HepG2 and Huh7 hepatocellular carcinoma cells (HCCs). MID1IP1 was highly expressed in HepG2, Huh7, SK-Hep1, PLC/PRF5, and immortalized hepatocyte LX-2 cells more than in normal hepatocyte AML-12 cells. MID1IP1 depletion reduced the viability and the number of colonies and also increased sub G1 population and the number of TUNEL-positive cells in HepG2 and Huh7 cells. Consistently, MID1IP1 depletion attenuated pro-poly (ADP-ribose) polymerase (pro-PARP), c-Myc and activated p21, while MID1IP1 overexpression activated c-Myc and reduced p21. Furthermore, MID1IP1 depletion synergistically attenuated c-Myc stability in HepG2 and Huh7 cells. Of note, MID1IP1 depletion upregulated the expression of ribosomal protein L5 or L11, while loss of L5 or L11 rescued c-Myc in MID1IP1 depleted HepG2 and Huh7 cells. Interestingly, tissue array showed that the overexpression of MID1IP1 was colocalized with c-Myc in human HCC tissues, which was verified in HepG2 and Huh7 cells by Immunofluorescence. Notably, depletion of CCR4-NOT2 (CNOT2) with adipogenic activity enhanced the antitumor effect of MID1IP1 depletion to reduce c-Myc, procaspase 3 and pro-PARP in HepG2, Huh7 and HCT116 cells. Overall, these findings provide novel insight that MID1IP1 promotes the growth of liver cancer via colocalization with c-Myc mediated by ribosomal proteins L5 and L11 and CNOT2 as a potent oncogenic molecule.

## 1. Introduction

Though hepatocellular carcinoma (HCC) is reported to be the sixth most common cancer in the world according 2018 Global Cancer Statistics [1], more effective therapies are still being required compared to classical treatments such as chemotherapy with a kinase inhibitor sorafenib [2,3], surgery [4], immunotherapy [1] and liver transplantation [5]. Recently, multi-target therapy has been attractive in HCC treatment [5,6]. Accumulating evidence reveals that HCC progression is closely associated with epithelial-mesenchymal transition (EMT), tumor microenvironment, tumor–stromal interactions, senescence bypass and cancer stem cells [7].

Among oncogenic molecules, Myc is elevated or dysregulated in approximately 70% of human cancers, including HCC [8,9]. Indeed, the *MYC* oncogene family comprising of *C-MYC*, *MYCN* and *MYCL* encode c-Myc, N-Myc and L-Myc, which are involved in ribosome biogenesis, cell-cycle progression, protein translation and metabolism, with a variety of biological functions including proliferation, differentiation, survival and immune surveillance [10]. Also, Myc is known to regulate ribosome biogenesis and protein synthesis through the transcriptional control of RNA and protein components of ribosomes [11].

It is well documented that ribosomal RNA (rRNA) is transcribed from ribosomal DNA (rDNA) to bind to ribosomal proteins, which have the small subunit consisting of a single rRNA chain and 33 ribosomal proteins, and the large subunit including three rRNA chains and 47 ribosomal protein large subunits (RPLs) in humans [12,13]. Also, emerging evidence reveals that ribosomal protein mutations are critically involved in ribosomopathies and carcinogenesis [14], since ribosomal proteins such as L5, L11, L18 and L29 are influenced by oncogenic factors and dysregulated translational proteins [15,16,17,18].

Interestingly, midline1 interacting protein 1 (MID1IP1), one of the glucose-responsive genes regulated by carbohydrate-responsive element-binding protein (ChREBP) [19], is known to act as a negative regulator of AMP-activated protein kinase (AMPK) in lipid metabolism [20]. Similarly, CCR4-NOT2 (CNOT2) is reported to promote lipid metabolism [21], angiogenesis [22], proliferation [23] and autophagy [24] as a potent oncogenic molecule.

Thus, in the present study, considering that cancer cells favor metabolism through glycolysis rather than efficient oxidative phosphorylation [25,26], the underlying oncogenic potential of MID1IP1 was explored in HCC growth in association with c-Myc signaling mediated by ribosomal protein L5 or L11 and CNOT2 in HCC cells and tissues.

## 2. Materials and Methods

### 2.1. Cell Culture

Hepatocellular carcinoma cell lines such as HepG2 (American Type Culture Collection (ATCC)^®^ HB-8065^™^), Hep3B (ATCC^®^ HB-8064), Huh7 (PTA-4583), PLC/PRF5 (ATCC^®^ CRL-8024^™^), SK-Hep1 (ATCC^®^ HTB-52^™^), Chang human liver cells (ATCC^®^ CCL-13^™^), AML-12 mouse hepatocytes (ATCC^®^ CRL2254™), LX-2 human hepatic stellate cells (SCC064, Sigma-Aldrich, St. Louis, MO, USA) and human colorectal cancer HCT116 (ATCC^®^ CCL-247^™^) were used in this study. HepG2 cells were cultured in Modified Eagle’s medium (MEM, catalog NO. LM 007-54, WelGENE, Gyeongsan, Korea). Hep3B cells were cultured in Dulbecco Modified Eagle’s medium (DMEM, catalog NO. LM 001-05, WelGENE, Gyeongsan, Korea). Huh7 and PLC/PRF5 cells were cultured in Roswell Park Memorial Institute 1640 (RPMI, catalog NO. LM 011-01, WelGENE, Gyeongsan, Korea). All cells were cultured in the aforementioned medium supplemented with 10% fetal bovine serum (FBS) and 1% antibiotic solution (100 units/mL penicillin and 100 µg/mL streptomycin) at 37 °C with 5% CO_2_.

### 2.2. RNA Interference

HepG2 and Huh7 cells were seeded onto culture plates overnight and transfected with the mixtures of MID1IP1 siRNA oligonucleotides (sense: 3′-CACCUUCUUCGACCCAUCU(dtdt) and antisense: 5′-AGAUGGGUCGAAGAAGGUG(dtdt) (Bioneer, Daejeon, Korea)) or scramble siRNA control (Cat.No.SN-1003) purchased from Bioneer (Bioneer, Daejeon, Korea), and CNOT2 siRNA(SC-72937), L5 siRNA (SC-78649), L11 siRNA (SC-60076) or scramble siRNA control purchased from Santa Cruz Biotechnology (Dallas, TX, USA), which were adjusted at 40 nM by using transfection reagent (INTERFERin, Polyplus, France) according to the manufacturer’s protocols. The transfected cells were incubated for 60–72 h for the next experiment.

### 2.3. Cytotoxicity Assay

HepG2 and Huh7 cells transfected with MID1IP1 siRNA or scramble siRNA control were seeded into a 96-well plate at a density of 7 × 10^3^ cells/well and incubated overnight at 37 °C for 72 h. Then, 30 μL of MTT (3-(4,5-dimethylthiazol-2-yl)-2,5-diphenyltetrazolium bromide; 1 mg/mL, Merck KGaA, Darmstadt, Germany) was distributed to each well of the plate and incubated for 2 h at 37 °C in the dark. The supernatant was carefully aspirated and 100 μL of dimethyl sulfoxide (DMSO) (Ducksan, Korea) was added and the optical density values were measured in a Biorad microplate reader model 680 (Biorad, Hercules, CA, USA) at 570 nm.

### 2.4. Colony Formation Assay

HepG2, Huh7 and Hep3B cells transfected with MID1IP1 siRNA or scramble siRNA control were distributed onto a 12-well cell culture plate at 1 × 10^3^ cells/well and incubated for 1 week to form colonies at 37 °C with 5% CO_2_. The colonies were stained with Diff Quick solution 2 (Cat No.38721, Sysmex corporation, Kobe, Hyogo, Japan), dried overnight and counted.

### 2.5. Cell Cycle Analysis

Based on Jung et al.’s paper [27], briefly, HepG2 and Huh7 cells transfected with MID1IP1 siRNA or scramble siRNA control were cultured to confluency for 72 h and then were harvested by using a Becton Dickinson (BD) falcon cell scraper. The detached cells were washed twice with cold PBS and fixed in 75% ethanol at −20 °C for overnight. The cells were incubated with RNase A (10 mg/mL) for 1 h at 37 °C and then stained with propidium iodide (50 μg/mL) for 30 min. The stained cells were analyzed with FACSCalibur by using CellQuest Software version 5.2.1 (Becton Dickinson, Franklin Lakes, NJ, USA).

### 2.6. Terminal Deoxynucleotidyl Transferase-dT-Mediated dUTP Nick End Labelling (TUNEL) Assay

As shown in Jung et al.’s paper [28], HepG2 and Huh7 cells transfected with MID1IP1 siRNA or scramble siRNA control were cultured for 72 h, fixed in permeabilization solution and incubated with TUNEL assay mixture for 60 min. The TUNEL-stained cells were observed under a FLUOVIEW FV10i confocal microscope (Olympus, Tokyo, Japan).

### 2.7. Tissue Microarray and Immunohistochemistry

As shown in Jung et al.’s paper [27], HCC patient tissue microarray purchased from Biomax (BC03119b, USA) was subjected to immunohistochemistry (IHC) staining by using Discovery XT immunohistochemistry (IHC)/in situ hybridization (ISH) Research Platform (Roche, Mannheim, Germany). The microarray panel has 120 cases of cholangiocellular carcinoma (15 tissues), hepatocellular carcinoma (95 tissues) and matched normal tissues (10 tissues). These tissues were fixed with 4% paraformaldehyde, dehydrated, embedded in paraffin and sectioned by 4 µm. The slides were incubated with the antibodies of MID1IP1 (1:200) (ab224550, Abcam, Cambridge, UK) and c-Myc (ab32072, Abcam, Cambridge, UK) at 4 °C overnight. Sections were washed in PBS and incubated with secondary antibody biotinylated goat anti-rabbit (1:150; Vector laboratories, Burlingame, CA, USA) for 30 min. After further washes, the antibodies were detected with the Vector ABCcomplex/horseradish peroxidase kit (Vector Laboratories) and color developed with DAB (3,3′-diaminobenzidine tetrahydrochloride). Then, the stained slides were analyzed for staining efficiency (score 0 (no staining), 1 (<5% of neoplastic cells staining), 2 (5–10% of neoplastic cells staining), 3 (10–30% of neoplastic cells staining) or 4 (>50% of neoplastic cells staining)) under a light microscope. The correlation coefficient between MID1IP1 and c-Myc was determined with mean values for % of positively stained cells with their standard deviations by using Fisher’s exact *t*-test.

### 2.8. Western Blotting

HepG2, Huh7, Hep3B and HCT116 cells transfected with MID1IP1 siRNA or scramble siRNA control were lysed in radioimmunoprecipitation assay (RIPA) buffer (2 mM ethylenediaminetetraacetic acid (EDTA), 150 mM NaCl, 50 mM Tris-HCl and 1% Triton X-100) containing protease inhibitors and phosphatase inhibitors on ice using NP-40 buffer, and spun down for 20 min at 4 °C. The supernatants were quantified for protein concentration using *DC* Protein Assay (Bio-Rad Laboratories, Hercules, CA, USA) and the protein samples were separated on 4–12% NuPAGE Bis–Tris gels and transferred to a Hybond ECL transfer membrane for detection with antibodies for MID1IP1 (ProteinTech Antibody Group, Chicago, IL, USA), PARP, Caspase-3, p21, snail, CNOT2 (Cell Signaling Technology, Beverly, MA, USA), c-Myc (Abcam, Cambridge, UK), and β-actin (Sigma, St. Louis, MO, USA). These antibodies were diluted in 3% bovine serum albumin (BSA) and in PBS-Tween20 (1:500–1:2000) at 4 °C overnight, washed with PBS-Tween20 and then incubated with horseradish peroxidase (HRP)-conjugated secondary antibody (Santacruz, sc-516102, sc-2357) (1:2000). The protein expression was visualized by using ECL Western blotting detection reagent (GE Healthcare, Amersham, UK).

### 2.9. c-Myc Stability Assay Using Cycloheximide

Based on Lee et al.’s paper [29], HepG2 and Huh7 cells transfected with MID1IP1 siRNA or negative control siRNA were cultured with 50 μg/mL cycloheximide for various times (0, 15 and 60 min) and were subjected to Western blotting with antibodies of c-Myc, MID1IP1, β actin and/or snail, p21.

### 2.10. Immunofluorescence Assay

HepG2 and Huh7 cells transfected with MID1IP1 siRNA or negative control siRNA were fixed with 4% formaldehyde and were permeabilized in 0.1% Triton X-100, based on Lee et al.’s paper [29]. The fixed cells were washed and incubated with the specific antibodies of MID1IP1 (1:200; Invitrogen, Waltham, MA, USA) and c-Myc (1:500; Santa Cruz Biotechnology, Dallas, TX, USA) overnight at 4 °C and then with secondary antibodies of Alex Fluor 488 goat rabbit immunoglobin G (IgG) antibody (1:500) (Invitrogen, Waltham, MA, USA) and Alexa Fluor 546 goat mouse-IgG antibody (1:500) (Invitrogen, Waltham, MA, USA) for 2 h at room temperature. The nuclei of the cells were stained with 4,6-diamidino-2-phenylindole (DAPI) and then were observed by using a FLUOVIEW FV10i confocal microscope and also the images of MID1IP1 and c-Myc-stained cells were taken by a Delta Vision imaging system.

### 2.11. Statistical Analysis

All data were expressed as means ± standard deviation (SD). Student’s *t*-test was used for comparison of two groups. Also, the one-way analysis of variance (ANOVA) followed by Tukey’s post-hoc test was performed for multi-group comparison using GraphPad Prism software (Version 5.0, San Diego, CA, USA). For overall survival (OS), associations between MID1IP1 subtypes and survival rate were then calculated by Kaplan–Meier analysis using a log-rank test.

## 3. Results

### 3.1. Endogenous Expression Levels in a Variety of Cells and Effect of MID1IP1 Depletion on the Viability and the Number of Colonies in HCCs

To assess the effect of MID1IP1 depletion on the viability and proliferation of HCCs, MTT assay and colony formation assay were conducted in HCCs. MID1IP1 was highly expressed in HepG2, Huh7, SK-Hep1, PLC/PRF5 and immortalized hepatocyte LX-2 cells more than in normal hepatocyte AML-12 cells and Hep3B cells (Figure 1A). Also, depletion of MID1IP1 depletion reduced the viability of Huh7 and Hep3B cells in a time-dependent fashion compared to the untreated control (Figure 1B). Also, the number of colonies were significantly reduced in MID1IP1-depleted HepG2, Huh7 and Hep3B cells compared to the untreated control (Figure 1C).

### 3.2. MID1IP1 Depletion Increases Sub G1 Population and Increases the Number of TUNEL-Positive Cells in HepG2 and Huh7 Cells

To check whether the cytotoxicity by MID1IP1 depletion is due to apoptosis, cell cycle analysis and TUNEL assay were performed in HepG2 and Huh7 cells. Here, MID1IP1 depletion increased the sub G1 population (Figure 2A) and the number of TUNEL-positive cells (Figure 2B) in HepG2 and Huh7 cells compared to the untreated control.

### 3.3. MID1IP1 Depletion Attenuates the Expression of Pro-PARP, c-Myc and CNOT2 and Activates p21 in Cancers

Consistently, to confirm the effect of MID1IP1 depletion on apoptosis-related proteins, Western blotting was conducted in HCCs and HCT116 cells. Here, MID1IP1 depletion attenuated the expression of pro-PARP, MID1IP1 and c-Myc in HepG2 and Huh7 cells compared to the untreated control (Figure 3A). In contrast, MID1IP1 overexpression activated c-Myc and reduced p21 in HepG2 cells (Figure 3B). Also, MID1IP1 depletion cleaved PARP in Hep3B cells (Figure 3C). Similarly, MID1IP1 depletion attenuated the expression of c-Myc, pro-PARP and CNOT2 in HCT116 cells in a concentration-dependent fashion (Figure 3D).

### 3.4. MID1IP1 Depletion Attenuates c-Myc Stability in HepG2 and Huh7 Cells in the Presence of Cycloheximide

To confirm the effect of MID1IP1 depletion on c-Myc stability, DNA synthesis inhibitor cycloheximide assay was performed in HepG2 and Huh7 cells. Herein, MID1IP1 depletion synergistically suppressed c-Myc stability compared to the cycloheximide-alone control in Huh7 cells (Figure 4A). Likewise, MID1IP1 depletion synergistically reduced the expression of c-Myc, p21 and snail compared to the cycloheximide-alone control in HepG2 cells (Figure 4B).

### 3.5. Colocalization Between MID1IP1 and c-Myc in Human HCC Tissues, HepG2 and Huh7 Cells

It is well known that colocalization indicates the spatial overlap between two or more different fluorescent targets located in the same area of the cell [30]. Here, human HCC patient tissue microarray strongly demonstrated the close association between MID1IP1 and c-Myc, with a significant correlation coefficient (*p* = 0.004) by Fisher’s exact test, since MID1IP1 and c-Myc were coexpressed in the same area of HCC tissue (Figure 5A). Consistently, Immuoflourescence showed colocalization between MID1IP1 and c-Myc with a yellowish color when merging the fluorescence of green (MID1IP1) and red (c-Myc) (Figure 5B).

### 3.6. The Critical Role of Ribosomal Protein L5 or L11 in c-Myc Inhibition by MID1IP1 Depletion in HepG2 and Huh7 Cells

Given that ribosomal proteins such as L5, L11, L18 and L29 are known to be influenced by oncogenic factors and dysregulated translational proteins [15], the critical role of ribosomal protein L5 or L11 was explored in c-Myc inhibition by MID1IP1 depletion in HepG2 and Huh7 cells. Here, MID1IP1 depletion activated L5 and L11 and reduced c-Myc, while loss of L5 or L11 reversed the inhibitory effect of MID1IP1 depletion on c-Myc in HepG2 (Figure 6A,B) and Huh7 cells (Figure 6C,D).

### 3.7. CNOT2 Knockdown Enhances Antitumor Effect of MID1IP1 Depletion in HepG2, Huh7 and HCT116 Cells

To assess the important role of CNOT2 in MIDIIP1-mediated oncogenesis in HCCs, a CNOT2 siRNA study was conducted in HepG2 and Huh7 cells, since several oncogenic factors are orchestrated in carcinogenesis [31]. Here, CNOT2 knockdown enhanced the inhibitory effect of MID1IP1 depletion on c-Myc, pro-PARP and pro-casapse 3 in HepG2 cells, and also CNOT2 depletion promoted the inhibitory effect of MID1IP1 depletion on c-Myc, pro-PARP and pro-casapse3 in Huh7 cells (Figure 7A,B). Furthermore, CNOT2 knockdown enhanced the inhibitory effect of MID1IP1 depletion on c-Myc and pro-PARP, even in HCT116 colorectal cancer cells (Figure 7C).

## 4. Discussion

In the current work, the underlying molecular mechanism of MID1IP1 as a potent oncogene was elucidated in association with c-Myc mediated by ribosomal protein L5 or L11 and CNOT2 in HCCs, with the hypothesis that MID1IP1 with hyperlipidemic activity [19,20] may act as a potent oncogene in HCC progression. Herein, MID1IP1 was highly expressed in HepG2, Huh7, SK-Hep1, PLC/PRF5 and immortalized hepatocyte LX-2 cells more than in normal hepatocyte AML-12 cells and Hep3B cells, indicating endogenous overexpression of MID1IP1 in HCCs, except for p53-deficient Hep3B cells.

In the present study, MID1IP1 depletion showed cytotoxic and anti-proliferative effects and increased the sub G1 population and the number of TUNEL-positive cells in HepG2 and Huh7 cells, indicating that MID1IP1 is involved in the growth, proliferation and antiapoptosis of HCCs.

Accumulating evidence reveals that overabundance and mutations of the Myc family comprising of c-Myc, N-Myc and L-Myc are shown in several cancers [32]. Among many apoptosis-related proteins, proteolytic cleavages of PARP into 89 kDa and 24 kDa fragments by caspases are considered an early indicator of apoptosis [33] and caspase 3 is known as an executioner caspase in apoptosis by the destruction of cellular structures, including DNA fragmentation or cytoskeletal proteins [34]. p21, also termed p21^WAF1/Cip1^, is known as a key cell cycle regulator that arrests cells at the G1 and G2 phases and is also reported to trigger apoptosis [35]. Our Western blotting revealed that MID1IP1 depletion attenuated pro-PARP, c-Myc and activated p21 in HepG2 and Huh7 cells, while MID1IP1 overexpression activated c-Myc and reduced p21 in HepG2 cells, demonstrating the pivotal role of MID1IP1 in HCC progression. Furthermore, MID1IP1 depletion enhanced suppression of c-Myc stability in HepG2 and Huh7 cells in the presence of DNA synthesis inhibitor cycloheximide, implying that MID1IP1 regulates c-Myc stability.

It is well documented that Myc acts as a regulator of ribosome biogenesis and protein synthesis [11,36] and the ribosomal large subunit comprises three rRNA chains and 47 RPLs in humans [37].

Here, MID1IP1 depletion induced upregulation of ribosomal protein L5 or L11, whereas depletion of L5 or L11 activated c-Myc in MID1IP1-depleted HepG2 and Huh7 cells, implying that L5 or L11 as a tumor suppressor regulates c-Myc, which was supported by Liao et al.’s paper that ribosomal proteins L5 and L11 cooperatively inactivate c-Myc via RNA-induced silencing complex [17]. Consistently, human tissue array showed that the overexpression of MID1IP1 was colocalized with c-Myc in human HCC tissues, with a significant correlation coefficient by Fisher’s exact test and also, Immunofluorescence confirmed the yellowish color by merging green and red signals between MID1IP1 and c-Myc in HepG2 and Huh7 cells, strongly demonstrating colocalization between MID1IP1 and c-Myc. 

Emerging evidence reveals that CNOT2 acts as an oncogene [22,23], adipogenic molecule [21] and autophagy regulator [22,38]. Considering that nonalcoholic fatty liver disease (NAFLD) increases the risk of HCC progression [39], and obesity-like condition promotes the proliferation of breast cancer via Warburg effect inversion [40], hyperlipidemic or adipogenic molecules may be similarly involved in cancer progression. Indeed, CNOT2 depletion enhanced the antitumor effect of MID1IP1 depletion to reduce c-Myc, procaspase 3 and pro-PARP in MID1IP1-depleted HepG2, Huh7 and HCT116 cells, suggesting a close association between MID1IP and CNOT2. However, further study is required to evaluate the pivotal roles and protein–protein interaction (PPI) of MID1IP1 and CNOT2 in cancers via the Warburg effect in vitro and in vivo.

Taken together, these findings provide novel insight that MID1IP1 promotes the growth of liver cancer via colocalization with c-Myc mediated by ribosomal protein L5 and L11 and CNOT2 as a potent oncogenic molecule. 

## 5. Conclusions

Collectively, through our current study, MID1IP1 depletion reduced the viability and proliferation, and increased the sub G1 population and the number of TUNEL-positive cells in HepG2 and Huh7 cells. Consistently, MID1IP1 depletion attenuated pro-PARP, c-Myc and activated p21 and synergistically attenuated c-Myc stability by cycloheximide assay in HepG2 and Huh7 cells, while MID1IP1 overexpression activated c-Myc and reduced p21 in HepG2 cells. Notably, MID1IP1 depletion upregulated L5 or L11, while loss of L5 or L11 rescued c-Myc in MID1IP1-depleted HepG2 and Huh7 cells. Interestingly, MID1IP1 was colocalized with c-Myc in human HCC cells and human tissues by IHC and Immunofluorescence. Furthermore, CNOT2 depletion enhanced the antitumor effect of MID1IP1 depletion to reduce c-Myc, procaspase 3 and pro-PARP in MID1IP1-depleted HepG2, Huh7 and HCT116 cells. Overall, these findings suggest that MID1IP1 promotes liver cancer progression via colocalization with c-Myc mediated by ribosomal protein L5 and L11 and CNOT2 as a potent oncogenic molecule.

## Figures and Tables

**Figure 1 cells-09-00985-f001:**
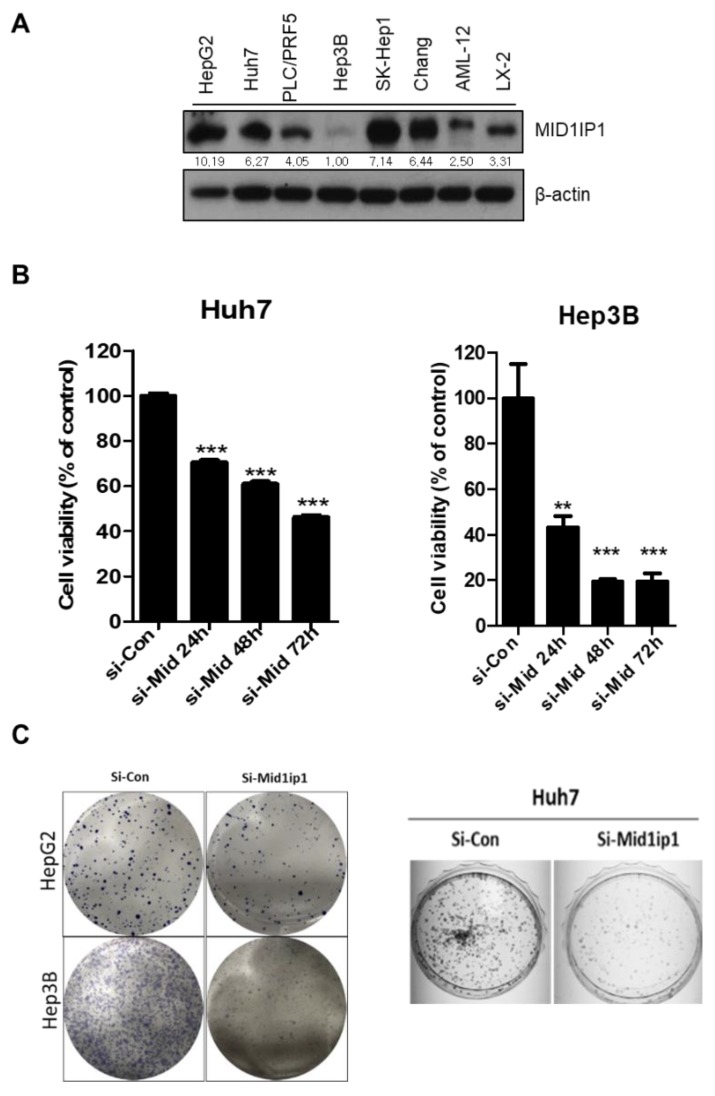
Endogenous expression levels of MID1IP1 in a variety of cells and effect of MID1IP1 depletion on the viability and the number of colonies in HCCs. (**A**) MID1IP1 protein level in various liver cancers and normal cells was detected by Western blotting. (**B**) Effect of MID1IP1 depletion on the viability of Huh7 and Hep3B cells. Huh7 and Hep3B cells transfected with MID1IP1 siRNA or negative control siRNA were incubated overnight at 37 °C for 0, 24, 48, 72 h and then exposed to 30 μL of MTT solution and the optical density values were measured in Biorad microplate reader model 680 (Biorad, USA) at 570 nm. (**C**) Effect of MID1IP1 depletion on the proliferation of HepG2 and Huh7 cells. HepG2, Huh7 and Hep3B cells transfected with MID1IP1 siRNA or negative control siRNA were incubated for 1 week and then the colonies were stained with Diff Quick solution 2 (Cat No.38721, Sysmex corporation, Japan), dried overnight and counted. ** *p* < 0.01, *** *p* < 0.001 vs. siRNA-Control. Values of Western blot images represent relative level of protein expression/β-actin.

**Figure 2 cells-09-00985-f002:**
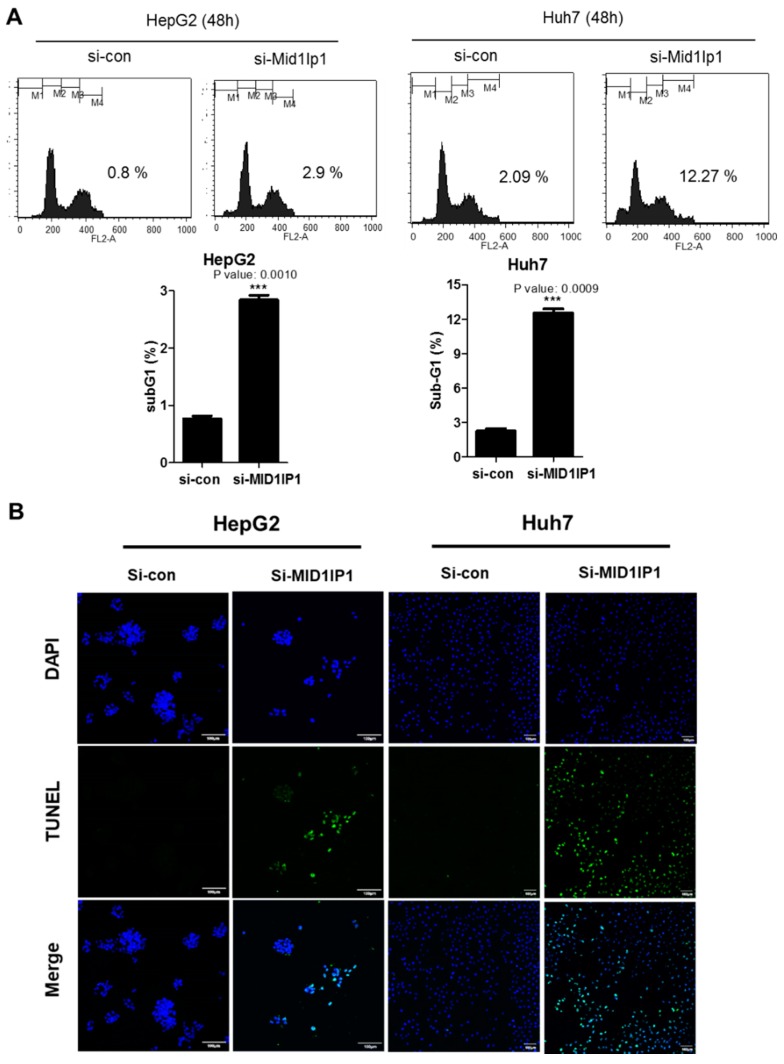
MID1IP1 depletion increases the sub G1 population and increases the number of TUNEL-positive cells in HepG2 and Huh7 cells. (**A**) Effect of MID1IP1 depletion on sub G1 population in HepG2 and Huh7 cells after 48 h culture by flow cytometric analysis. (**B**) Effect of MID1IP1 depletion on TUNEL-positive cells in HepG2 and Huh7 cells. HepG2 and Huh7 cells transfected with MID1IP1 siRNA plasmid or negative control siRNA were cultured for 72 h and then were fixed with 4% paraformaldehyde for 30 min, exposed to TUNEL assay mixture for 60 min and TUNEL-stained cells were visualized by a FLUOVIEW FV10i confocal microscopy (Olympus, Tokyo, Japan).

**Figure 3 cells-09-00985-f003:**
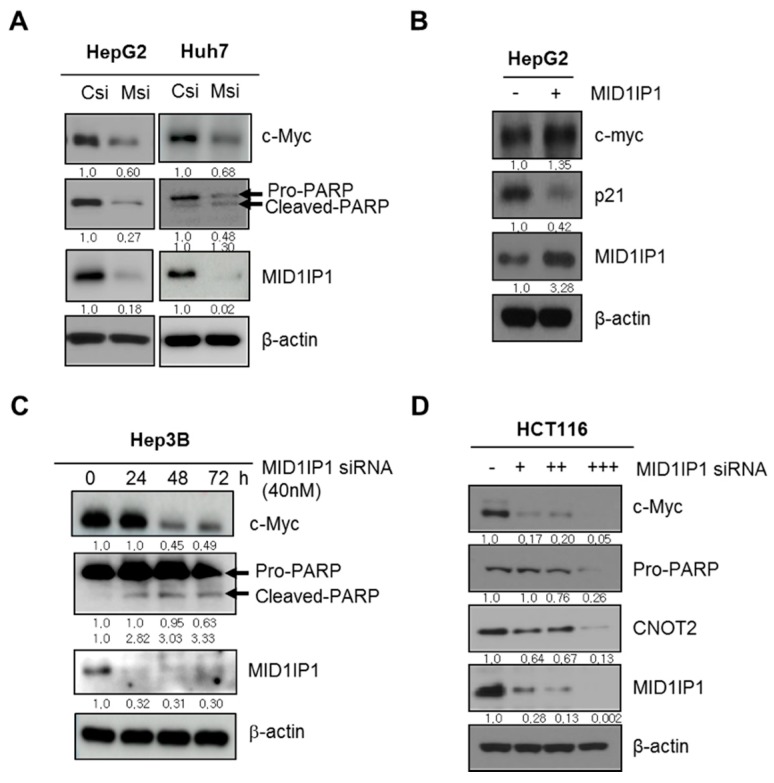
MID1IP1 depletion attenuates the expression of pro-PARP, c-Myc and CNOT2 and activates p21 in several cancers. (**A**) Effect of MID1IP1 depletion on pro-PARP (116 kDa), cleaved-PARP (89 kDa), MID1IP1 and c-Myc in HepG2 and Huh7 cells. HepG2 and Huh7 cells transfected with MID1IP1 siRNA (Msi) or negative control siRNA (Csi) were cultured for 72 h and then were subjected to Western blotting with antibodies of pro-PARP, MID1IP1, c-Myc and β-actin. (**B**) Effect of MID1IP1 overexpression on MID1IP1, c-Myc and p21 in HepG2 cells. HepG2 cells transfected with MID1IP1 overexpression plasmid (+) and pcDNA3.0 control vector (−) were cultured for 72 h and then were subjected to Western blotting with antibodies of MID1IP1, c-Myc and p21. (**C**) Effect of MID1IP1 depletion on MID1IP1, pro-PARP (116 kDa) and cleaved-PARP (89 kDa) in Hep3B cells. (**D**) Effect of MID1IP1 depletion on MID1IP1, c-Myc, pro-PARP and CNOT2 in HCT116 cells. Concentrations of MID1IP1 siRNA; + (20 nM), ++ (40 nM), +++ (80 nM). Values of Western blot images represent relative level of protein expression/β-actin. Western blotting was conducted in triplicates.

**Figure 4 cells-09-00985-f004:**
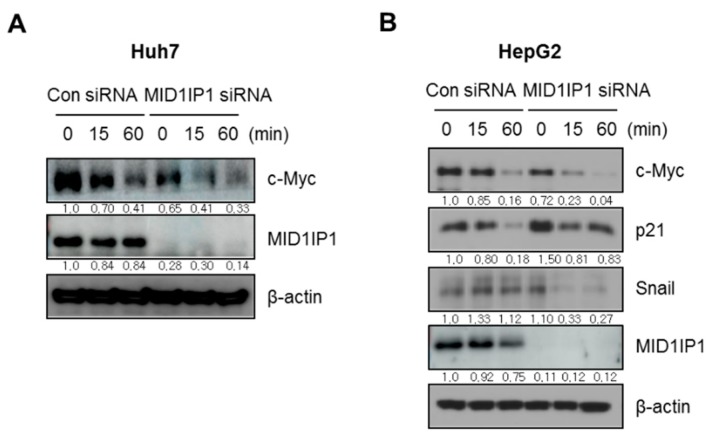
MID1IP1 depletion attenuates c-Myc stability in HepG2 and Huh7 cells in the presence of cycloheximide. (**A**) Effect of MID1IP1 depletion on c-Myc and MID1IP1 in Huh7 cells. Huh7 cells transfected with MID1IP1 siRNA or negative control siRNA were cultured with 50 μg/mL cycloheximide for 15 and 60 min and were subjected to Western blotting with antibodies of c-Myc, MID1IP1 and β actin. (**B**) Effect of MID1IP1 depletion on c-Myc, p21, snail and MID1IP1 in HepG2 cells. HepG2 cells transfected with MID1IP1 siRNA or negative control siRNA were cultured with 50 μg/mL cycloheximide for 15 and 60 min and were subjected to Western blotting with antibodies of c-Myc, MID1IP1, snail, p21 and β-actin. Values for Western blot images represent relative expression level of target protein/β-actin.

**Figure 5 cells-09-00985-f005:**
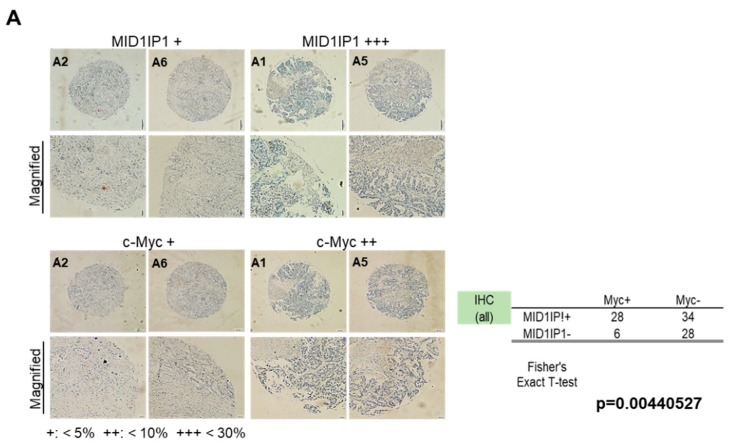
Colocalization between MID1IP1 and c-Myc in human HCC tissues, HepG2 and Huh7 cells. (**A**) Human HCC patient tissue microarray (two slides) purchased from Biomax (HLivH160CS01, Derwood, MD, USA) was subjected to immunohistochemistry (IHC) staining by Discovery XT IHC/ISH Research Platform (Roche, Mannheim, Germany). Expression level of MID1IP1 and c-Myc each of HCC tissues (62) and normal tissues (34) with correlation coefficient. (**B**) HepG2 and Huh7 cells transfected with MID1IP1 siRNA or negative control siRNA were cultured for 72 h and then were subjected to Immunoflourescence with antibodies of MID1IP1, c-Myc, Alex Fluor 488 goat rabbit IgG antibody (Invitrogen) and Alexa Fluor 546 goat mouse-IgG antibody and DAPI. ‘D’ indicates DAPI.

**Figure 6 cells-09-00985-f006:**
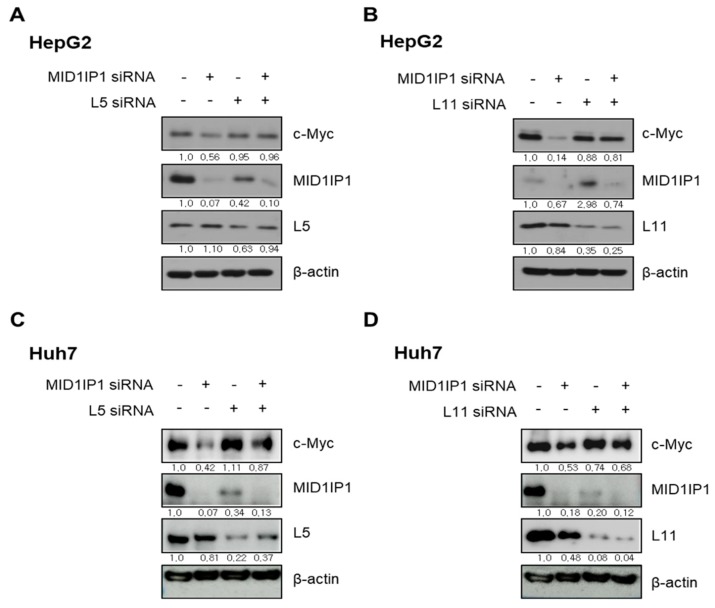
Effect of ribosomal protein L5 or L11 on c-Myc in MID1IP1 depleted HepG2 and Huh7 cells. (**A**) Effect of RPL5 depletion on c-Myc in MID1IP1 depleted HepG2 cells. (**B**) Effect of RPL11 depletion on c-Myc in MID1IP1 depleted HepG2 cells. (**C**) Effect of RPL5 depletion on c-Myc in MID1IP1 depleted HepG2 cells. (**D**) Effect of RPL11 depletion on c-Myc in MID1IP1 depleted HepG2 cells. −, untreated; +, treated. Values of Western blot images represent relative level of protein expression/β-actin.

**Figure 7 cells-09-00985-f007:**
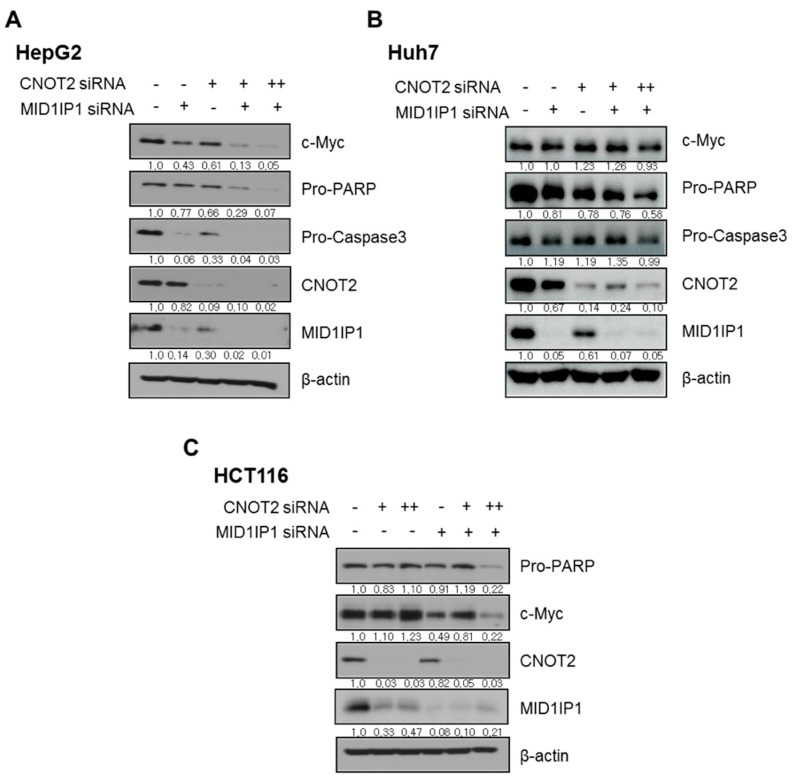
CNOT2 knockdown enhances antitumor effect of MID1IP1 depletion in HepG2, Huh7 and HCT116 cells. (**A**) Effect of CNOT2 knockdown on inhibitory effect of MID1IP1 depletion on c-Myc, pro-PARP and pro-Casapse3 in HepG2 cells. (**B**) Effect of CNOT2 knockdown on inhibitory effect of MID1IP1 depletion on c-Myc, pro-PARP and pro-Casapse3 in Huh7 cells. (**C**) Effect of CNOT2 knockdown on inhibitory effect of MID1IP1 depletion on c-Myc and pro-PARP in HCT116 cells. Concentration of CNOT2 siRNA; + (5 nM), ++ (10 nM), − untreated, and concentration of MID1IP1 siRNA; + (40 nM). Values of Western blot images represent relative level of protein expression/β-actin.

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
