# Peer review of "Colocalization of MID1IP1 and c-Myc is Critically Involved in Liver Cancer Growth via Regulation of Ribosomal Protein L5 and L11 and CNOT2"

_cells, 2020, doi:10.3390/cells9040985_

Round 1

Reviewer 1 Report

Thank you for addressing my concerns. 

Author Response

Thanks for your effort.

Reviewer 2 Report

The findings can be interesting. However, there are significant flaws that impair the scientific soundness and data quality that must be addressed.

Majors:

Improper controls for siRNA experiments: The manuscript includes many experiments using siRNA KD. However, proper controls were not set up. For instance, in Figure 1B, the authors stated that MID1IP1 KD using siRNA impair the viability of many HCC cell lines. However, their control is ‘untreated control’ as stated in Line 162. The impairment of cell viability could be due to the toxicity of transfection reagent, but not the effect of siRNA KD. Similar problems exist through out the manuscript. The authors are suggested to have complete control sets, ‘untreated control’, ‘scramble siRNA control’ (using their negative siRNA), and ‘mock control’ (with transfection reagent only). The data currently are not convincing throughout the manuscript due to the lack of appropriate controls.

Figure 2A: The authors will need replicates for the flow cytometry experiments and performed statistical analysis.

Figure 5A: Representative figures from tissue arrays are of poor quality. Even from the magnified images, we could not tell the colocalization.

Figure 5B: Even after revision, colocalization cannot be told from this. The authors need to pull out signal color channel for the lower panel for single cell. Please refer to Figure 5 from doi:10.1158/1535-7163.MCT-04-0253 for proper presentation of colocalization.

Line 323-325: There is an overstatement. The authors did not have data to show the correlation between MID1IP1 and HCC progression.

Minors:

Line 20: Please change the order of words. This current wording gives the false impression that both AML-12 and Hep3B were normal hepatocytes. Only AML-12 is normal hepatocyte.

Line 128: Should be ‘…were lysed in a lysis buffer…” and what buffer is it?

Line 127-133: For WB, how did you quantify the protein? BCA? Bradford? How much protein and antibody did you use? Please specify.

Author Response

The findings can be interesting. However, there are significant flaws that impair the scientific soundness and data quality that must be addressed.

Majors:

Improper controls for siRNA experiments: The manuscript includes many experiments using siRNA KD. However, proper controls were not set up. For instance, in Figure 1B, the authors stated that MID1IP1 KD using siRNA impair the viability of many HCC cell lines. However, their control is ‘untreated control’ as stated in Line 162. The impairment of cell viability could be due to the toxicity of transfection reagent, but not the effect of siRNA KD. Similar problems exist throughout the manuscript. The authors are suggested to have complete control sets, ‘untreated control’, ‘scramble siRNA control’ (using their negative siRNA), and ‘mock control’ (with transfection reagent only). The data currently are not convincing throughout the manuscript due to the lack of appropriate controls.

(Response) Thanks. Corrected as suggested.

Figure 2A: The authors will need replicates for the flow cytometry experiments and performed statistical analysis.

(Response)Thanks, We performed statistical analysis for Figure2A.

Figure 5A: Representative figures from tissue arrays are of poor quality. Even from the magnified images, we could not tell the colocalization.

(Response)Thanks for critical comments. However, despite poor quality of IHC staining, we can determine colocolization especially between two set of A1 and A5. Also, since weak evidence can be questioned by readers for colocalization between MID1IP1 and c-Myc, we performed immunofluorescence in Figure 5B. Here we can confirm colocalization between MID1IP1 and c-Myc.

Figure 5B: Even after revision, colocalization cannot be told from this. The authors need to pull out signal color channel for the lower panel for single cell. Please refer to Figure 5 from doi:10.1158/1535-7163.MCT-04-0253 for proper presentation of colocalization.

(Response) Thanks. We added Immunofluorescence picture in a single cell as shown in MCT-04-0253 article.

Line 323-325: There is an overstatement. The authors did not have data to show the correlation between MID1IP1 and HCC progression.

(Response)Thanks. Corrected as “ MID1IP1 promotes the growth of liver cancer.”

Minors:

Line 20: Please change the order of words. This current wording gives the false impression that both AML-12 and Hep3B were normal hepatocytes. Only AML-12 is normal hepatocyte.

(Response) Thanks. Corrected.

Line 128: Should be ‘…were lysed in a lysis buffer…” and what buffer is it?

(Response)Added in 2.8 Western blotting.

Line 127-133: For WB, how did you quantify the protein? BCA? Bradford? How much protein and antibody did you use? Please specify.

(Response) Thanks. Detailed information was added in Western blotting.

Reviewer 3 Report

In general, the manuscript is well written and the contents support the conclusion.

Most of data seems so clear and valuable. Since there are not many studies of MID1IP1, this study has a novelty. Therefore, I agree this manuscript can be published to represent a sufficient advance for consideration in Cells.

I have some comments, and I believe they might be helpful to the authors to increase its novelty.

  1. It requires English editing.
  • Line 15: was known one of -> ‘known as one of the glucose…’
  • Line 17: was never explored so far. Thus, -> was never explored. Thus,…
  • Line 19: PLC/PRF5, immortalized -> PLC/PRF5, and immortalized
  • Line 20 – 23: It is hard to understand this long sentence.
  • Abstract has too many Conjunctions.
  • Line 64: c-MyC -> c-Myc
  • Line 68-71: Use the same format of ®.
  • Line 77: 37℃ 5% CO2. -> 37℃ with 5% CO2.
  • Line 100-101: You should add the procedure of harvesting the cells.
  • Line 102-103: The sentence is a little bit awkward.
  • Line 106: 72h, were fixed -> 72h, fixed
  • Line 108: You should add the company and area for the FLUOVIEW FV10i.
  • Line 127: You missed the quotation of the paper (Koo’s).
  • Line 128: were lysed in a lysis on ice -> were lysed on ice
  • Line 136: Add space between ‘15’ and ‘and’.
  • Line 146: Add space between ‘2 h’ and ‘at’.
  • Line 179-180: This sentence doesn’t make sense (… were performed increases…)
  • Line 320: MID1IP -> MID1IP1

  1. MID1IP1 needs more explanation in Introduction.
  2. How many times did you perform the experiments for Fig. 1 & Fig. 2?
  3. The expression level of c-Myc is required in Fig. 3C because this WB was performed in Hep3B cells.
  4. The β-actin WB band looks weird in Fig. 3C.
  5. Why did you use HCT116 cells in Fig. 3D?
  6. Why did you show different sites for different antibodies in the same sample in magnified pictures in Fig. 5A?
  7. Fig 5. needs scale bar values in the Figure legend.
  8. In Fig. 6, when MID1IP1 is downregulated by the siRNA transfection, L11 looks not activated in HepG2 cells. Likewise, in Huh7 cells, L5 and L11 are not upregulated after MID1IP1 knockdown. But you mentioned ribosomal proteins are activated after MID1IP1 depletion.
  9. In Fig.7C, is MID1IP1 upregulated after CNOT2 depletion?

Author Response

Comments and Suggestions for Authors

In general, the manuscript is well written and the contents support the conclusion.

Most of data seems so clear and valuable. Since there are not many studies of MID1IP1, this study has a novelty. Therefore, I agree this manuscript can be published to represent a sufficient advance for consideration in Cells.

I have some comments, and I believe they might be helpful to the authors to increase its novelty.

  1. It requires English editing.
  • Line 15: was known one of -> ‘known as one of the glucose…’
  • (Response) corrected.
  • Line 17: was never explored so far. Thus, -> was never explored. Thus,…(Response) corrected.
  • Line 19: PLC/PRF5, immortalized -> PLC/PRF5, and immortalized(Response) corrected.
  • Line 20 – 23: It is hard to understand this long sentence.
  • (Response) corrected.
  • Abstract has too many Conjunctions.
  • (Response) Thanks..Corrected.
  • Line 64: c-MyC -> c-Myc.
  • (Response) corrected.
  • Line 68-71: Use the same format of ®. .
  • (Response) corrected, but LX-2 human hepatic stellate cells were purchased from Sigma-Aldrich, not ATCC.
  • Line 77: 37℃ 5% CO2. -> 37℃ with 5% CO2.
  • (Response) corrected.
  • Line 100-101: You should add the procedure of harvesting the cells. (Response) Added.
  • Line 102-103: The sentence is a little bit awkward.
  • (Response) corrected.
  • Line 106: 72h, were fixed -> 72h, fixed
  • (Response) corrected.
  • Line 108: You should add the company and area for the FLUOVIEW FV10i. (Response) Added.
  • Line 127: You missed the quotation of the paper (Koo’s).
  • (Response) Corrected.
  • Line 128: were lysed in a lysis on ice -> were lysed on ice
  • (Response) Corrected.
  • Line 136: Add space between ‘15’ and ‘and’.
  • (Response) Corrected.
  • Line 146: Add space between ‘2 h’ and ‘at’.
  • (Response) Corrected.
  • Line 179-180: This sentence doesn’t make sense (… were performed increases…)
  • (Response) Corrected.
  • Line 320: MID1IP -> MID1IP1
  • (Response) Corrected.
  • MID1IP1 needs more explanation in Introduction. (Response) Added.
  • How many times did you perform the experiments for Fig. 1 & Fig. 2?(Response) Thanks. All experiments were carried out in triplicates.
  • The expression level of c-Myc is required in Fig. 3C because this WB was performed in Hep3B cells. (Response) Thanks. We added now blot for c-Myc in Fig3c as requested.
  • The β-actin WB band looks weird in Fig. 3C. (Response)Thanks. Replaced by new blot.
  • Why did you use HCT116 cells in Fig. 3D? (Response) We wanted to confirm the effect of MID1IP1 depletion in HCT116 cells different from three HCC cell lines.
  • Why did you show different sites for different antibodies in the same sample in magnified pictures in Fig. 5A? (Response) Thanks. We checked colocalization between c-Myc and MID1IP1 at same sites in two set slide of same human tissues purchased.
  • Fig 5. needs scale bar values in the Figure legend. (Response)Thanks. Added.
  • In Fig. 6, when MID1IP1 is downregulated by the siRNA transfection, L11 looks not activated in HepG2 cells. Likewise, in Huh7 cells, L5 and L11 are not upregulated after MID1IP1 knockdown. But you mentioned ribosomal proteins are activated after MID1IP1 depletion. (Response)Sorry for making you confused. Corrected.
  • In Fig.7C, is MID1IP1 upregulated after CNOT2 depletion? (Response)Thanks for your critical comments. We repeated Western blotting and added new blot for MID1IP1 In Fig7c.

Round 2

Reviewer 2 Report

My concerns were addressed by the authors.

This manuscript is a resubmission of an earlier submission. The following is a list of the peer review reports and author responses from that submission.

Round 1

Reviewer 1 Report

The research article “Colocalization of MIDIIPI and c-Myc is critically involved in liver cancer progression via regulation of ribosomal protein L5 and L11 and CNOT2” attempts to explore the role of MID1IP1 in HCC. Although there are a couple of interesting findings in the study, the paper is seriously flawed in experimental design, result analysis and interpretation, and reporting.

Major concerns:

The quality of English is poor throughout the paper, starting from the first line of the abstract (line 12). Abstract (Line 16/17) “MID1IP1 was highly expressed in HepG2, Huh7, SK-Hep1, PLC/PRF5, immortalized hepatocyte LX-2 cells more than in normal hepatocyte AML-12 cells, 3T3L1 cells and Hep3B cells”…..no study comparing the expression level of MID1IP1 has been reported. None of the western blot images have been quantified, bands look weird and out of place. Just showing the blots and claiming that the level of expression has been altered is probably not scientific. And given that most of the authors claims rely on the western blots, the results do not justify the conclusions. Naming and notations are not uniform and not well-defined (e.g. +, ++, +++) Information regarding how many replicates were performed is lacking in the figure legend. 3A: In HepG2, there are clearly 2 bands of MID1IP1, this band disappears in HuH7. In Fig 4B, only one band is seen in the same cells. Is it a different isoform? Does the siRNA target one isoform and not the other? No explanation or discussion has been provided. Similarly, lane of MID1IP1 in siRNA treated HuH7 cells seems dirty with shades of band. In Huh7, it is difficult to see if the c-myc protein band is lower in treatment vs control siRNA group at all. Similar discordant results and extra band is seen for Pro-PARP and no explanation has been provided. Fig 3C for b-actin seems to have been cut. Lower band seems to be appearing for PARP (and not Pro-PARP as in Fig 3A) while no depletion of upper band is visible as claimed. Figure 4 lacks CHX only control. Each of the figure and sub-figures have similar issues. Figure 5 is probably the only novel and interesting finding showing colocalization between c-Myc and MID1IP1 which is by itself not enough to claim a role of the protein in HCC progression.

Reviewer 2 Report

Majors: 1. Line 36 – 66: The authors simply listed different proteins involved in the manuscript. Transition and logical order is needed to organize the introduction for the audience to better understand. 2. Line 154 – 158: The authors are supposed to show the knockdown efficiency of MID1IP1 using qPCR. 3. WB: The authors should attach all the uncut WB images in supplementary figures. 4. Figure 5A: How are we going to tell the co-expression of MID1IP1 and c-Myc in the same area of HCC tissue based on the provided figure? Besides, how did the authors get the correlation coefficient? 5. Figure 5B: The authors need to show figures with higher magnification. The current ones are not enough to show the subcellular colocalization. Even DAPI overlaps with both targets. 6. Why the authors chose L5 and L11 only but not L18 or L29? 7. Line 247: What led the authors to study the role of CNOT2? The manuscript needs to be more organized for the readers to follow the rationale of the study. 8. Extensive language editing is needed. Minors: 1. Line 69 – 72: Please specify the sources of the cells. 2. Line 82 - 83: Please provide the sequence of your negative control siRNA. 3. Line 118 – 124: Info for antibodies (c-Myc, AF-488 secondary) is missing. 4. Figure 6 has no figure title or legend.

Reviewer 3 Report

The present work focuses on elucidating the mechanism of action of MID1IP1 in liver cancer cells. Using multiple hepatocellular carcinoma cell lines, the authors report that MID1IP1 promotes liver cancer progression via co-localization with c-Myc mediated by L5, L11 and CNOT1. It is a well-designed and well conducted work. I have some queries and I would like the authors to provide appropriate explanation for that.

For colony formation assay: Has the authors treated with siRNA continuously for 2 weeks or the treatment is done only at the start of the assay. I think that the siRNA effect would not last for 2 weeks.

Please clarify what is the meaning of Csi and Msi in figure 3A.

Figure 3B authors introduced p21. What was the effect of MID1IP1 silencing on p21? Please provide a blot for p21 in absence of MID1IP1.

What do the +, ++, and +++ signify in the figure 3D? Is it the increasing concentration of siRNA or increasing time period of siRNA or something else?

In figure 3A and 4B, MID1IP1 immunoblot in Hepg2 cell lysate shows a double band in vehicle as well as in siRNA, but in figure 3B, 6C, and 6D, the MID1IP1 immunoblot in HepG2 shows a single band vehicle or overexpression. Can the authors explain this?

Line 16: Typo error.

Somewhere its Huh7 and somewhere it is mentioned as HuH7 (Eg. Line 15 it is written as Huh7 while line 20 it is mentioned as HuH7). Please correct it.

Line 37: Grammatical error.

Line 57: Grammatical error.

Line 61 to 64: It is a very long and confusing. Please convert it into multiple reader friendly sentences.

The manuscript must be thoroughly corrected by a native English speaker or professional English editors.

The figure legend for figure 6 is missing.

Reviewer 4 Report

The manuscript titled "Colocalization of MID1IP1 and c-Myc is critically involved in liver cancer progression via regulation of ribosomal protein L5 and L11 and CNOT2" by Ji Hoon Jung et al was really interesting, overall the study design.

In the title, MIDIIP1 changed to MID1IP1.